# Monitoring of the Endangered Cave Salamander *Speleomantes sarrabusensis*

**DOI:** 10.3390/ani13030391

**Published:** 2023-01-24

**Authors:** Roberto Cogoni, Milos Di Gregorio, Fabio Cianferoni, Enrico Lunghi

**Affiliations:** 1Unione Speleologica Cagliaritana, Quartu Sant’Elena, 09045 Cagliari, Italy; 2Dipartimento di Biologia, Università degli Studi di Pisa, 56126 Pisa, Italy; 3Istituto di Ricerca sugli Ecosistemi Terrestri (IRET), Consiglio Nazionale delle Ricerche (CNR), Sesto Fiorentino, 50019 Firenze, Italy; 4Museo di Storia Naturale dell’Università degli Studi di Firenze, “La Specola”, 50125 Firenze, Italy; 5Dipartimento di Medicina Clinica, Sanità Pubblica, Scienze Della Vita e dell’ambiente (MeSVA), Universià degli Studi dell’Aquila, 67100 L’Aquila, Italy; 6Associazione Natural Oasis, 59100 Prato, Italy; 7Unione Speleologica Calenzano, 50041 Calenzano, Italy

**Keywords:** *Hydromantes*, Plethodontidae, abundance, wildlife, amphibian, conservation

## Abstract

**Simple Summary:**

Here, we provide the information derived from the first monitoring activities performed on the endangered Sette Fratelli cave salamander, *Speleomantes sarrabusensis*. Adopting two different monitoring schemes, we estimated the abundance of four populations of *S. sarrabusensis*, providing important data for future status assessments of the species.

**Abstract:**

In this study, we performed the first monitoring activities on one of the most endangered amphibians in Europe, the Sette Fratelli cave salamander *Speleomantes sarrabusensis*. The data presented here are derived from two monitoring activities aiming to assess the status and abundance of four populations of *S. sarrabusensis*. With the first monitoring, we surveyed the well-known population occurring within artificial springs during the period 2015–2018, providing monthly data on the number of active individuals. With the second monitoring performed during spring to early summer of 2022, we surveyed four populations at three time points (the one from artificial springs and three from forested areas) and we provided the first estimation of the populations’ abundance. Furthermore, we analyzed for the first time the stomach contents from a population of *S. sarrabusensis* only occurring in forested environments. With our study, we provided the first information on the abundance of different populations of *S. sarrabusensis*, representing the starting point for future status assessments for this endangered species.

## 1. Introduction

The Sette Fratelli cave salamander, *Speleomantes sarrabusensis* Lanza et al., 2001 (Figure 1), is one of the European amphibian species with the smallest distribution (≤70 km^2^); it can be found only in the most south-eastern area of Sardinia (Italy) [1].

Before 2001, *S. sarrabusensis* was considered a sub-species of *S. imperialis* [1]. Although called “cave salamanders”, all the European plethodontid salamanders of the genus *Speleomantes* are epigean species that occur in caves and other subterranean environments to avoid unsuitable climatic conditions [2,3]. However, *S. sarrabusensis* is distributed over a granitic area where no caves exist, and thus it can be found only in surface environments or within anthropic structures characterised by a suitable inner microclimate [1,4]. The very small distribution and the lack of potential subterranean refuges are within the characteristics that make *S. sarrabusensis* highly sensitive to extinction risk [5].

The Sette Fratelli cave salamander is the *Speleomantes* species for which the available knowledge is the poorest and the most anecdotal. For example, observing captive individuals, Lanza and Leo [6] hypothesised that the species might be viviparous; however, recent studies confirmed its oviparity as for the rest of the genus [7]. Most of our lack of knowledge is probably due to the difficulties in observing the species when it hides in its refuges. Subterranean *Speleomantes* usually show a high detection probability, as individuals can be easily observed clinging onto cave walls [8,9,10]. On the contrary, detecting *Speleomantes* in a forested area is more challenging as individuals have plenty of places to hide, and they can be confused with the background [11]. The only well-known population of *S. sarrabusensis* is the one that occurs inside the artificial springs located on the Sette Fratelli Mountain (squared structures made of concrete with water tanks that cover about 80% of the inner surface; Figure 2) [1], where salamanders cling onto the flat walls and experience microclimatic conditions that are comparable with those found in subterranean environments [4]. Consequently, our knowledge of *S. sarrabusensis* is only based on studies performed on the population from artificial springs [7,12,13,14].

The present study provides new information on the distribution and life history of the strictly protected *S. sarrabusensis*. Specifically, we provide data that originates from two different monitoring schemes. The first was long-term monitoring which was interested in the well-known artificial springs aiming to estimate the population trend over a period of four years [15]. Although in this part of the study we provide the number of individuals observed within each spring, we should remark that the two structures are not far enough from each other to assume that they hold two independent populations [16]. In the second part, we performed repeated surveys in both the artificial springs (considered as a single population) and in three different forested areas to provide the first estimation of their abundance (i.e., the abundance of active individuals at that time). The occurrence of these three independent epigean populations was previously assessed from literature and due to novel discoveries. The need for improving the information on this poorly known species was recently boosted by a large death toll observed in the summer of 2021, where 14 individuals were found dead. We still do not know the cause of such a death toll, but this dramatic event was one of the reasons behind the change in the species conservation status to critically endangered [5].

## 2. Materials and Methods

The first long-term monitoring of the artificial springs (Figure 2A,B) involved monthly surveys from March 2015 until June 2018. During each survey, one operator counted the number of active individuals of *S. sarrabusensis* which are clearly visible clinging onto the flat concrete walls (Figure 2C).

At the end of the survey, at about 5 m from the entrance, air temperature and humidity were recorded at the ground level using a thermohygrometer (accuracy ±0.5 °C, ±2.5%). We used Linear Models (LM) [17] to evaluate the potential variation of the number of observed individuals during the period of monitoring. The number of observed individuals was used as the dependent variable, the month and the year as fixed factors, and the spring identity as a random factor. The likelihood ratio test was used to test the significance of variables.

The second monitoring occurred in 2022. Besides the well-known population from the artificial springs, we surveyed three additional epigean populations aiming to perform the first estimation of their abundance and to provide much-improved information on species status. Two sites were characterized by forested areas mainly composed of evergreen oak *Quercus ilex* L., 1753 and the strawberry tree *Arbutus unedo* L., 1753, while one was characterized by Mediterranean scrub. The exact coordinates of the populations are not provided due to conservation concerns [18]. The three epigean populations were surveyed in March, while the one occurring inside the artificial springs was surveyed in June. These periods were chosen based on the general knowledge of *Speleomantes* phenology, where individuals are more active in surface environments during periods characterized by higher precipitation and relatively low temperatures, while during hot and dry periods the subterranean abundance should be the highest [9,11,19]; see also Table 1. Each population was surveyed three times within a period of 30 days, to guarantee population closure and to avoid strong microclimatic fluctuations [9,20]. During each survey, individuals were searched within the defined area and counted without providing any further disturbance. In each site, we selected a defined area to monitor according to the local characteristics of the territory. For the first site (forest #1) we selected a transect of 962 m, for the second site (forest #2) we selected a transect of 219 m; both transects were defined using GPS tracks. In the third site (scrub #1) we defined a plot of 320 m^2^ in which we adopted a serpentine (bustrophedic) search of salamanders on the ground and under the stones, avoiding inaccessible dense bushes. The overall surveyed area of the artificial springs is 70 m^2^ of building footprint (about 4 × 7.5 m the first and 4 × 10 m the second). Inside the artificial springs, a single operator searched for *Speleomantes* on the flat walls, with a constant sampling effort of 7.5 min per 3 linear meters [19]. For the epigean populations, two operators simultaneously searched for individuals on the ground (also lifting logs/stones) and on the trees [21] within the defined area. The two forested areas were surveyed with an average sampling effort of 1 h/1270 m^2^ per person (1 h/1227 m^2^ for the first and 1 h/1314 m^2^ for the second site), while for the scrub we dedicated 1 h for the whole plot (320 m^2^). During the third survey on each site, we measured temperature and humidity at the ground with a thermohygrometer (accuracy ±0.5 °C, ±2.5%). Estimations of the populations’ abundance were performed using *N*-mixture models [22].

Considering the large number of individuals observed from site forest #1 (see Results), we selected this population to perform an analysis on the stomach contents of individuals. During the last survey, individuals were captured and then we proceeded as follows: each individual was weighed using a digital scale (0.01 g), photographed for post hoc estimation of both their snout-vent length (SVL) and total length (TL) both SVL and TL; [23,24], and stomach flushed to collect the residuals of their last foraging activity [25]. Adult salamanders showing male’s sexual characteristics (mental chin, conical shape of the head) were considered males [1]. The SVL was then used to distinguish between adults females (≥55 mm) and juveniles (<55 mm) [7]. Individuals were then stabled in situ inside fauna boxes; this allowed us to perform a second round of capture–photograph–stomach-flushing the following night, without the risk of recapturing the same individuals. The recognised prey items were counted and divided following Lunghi, et al. [26]. In brief, consumed prey were divided according to their taxonomic order and possibly also according to their life stage (larva vs. adult). In two circumstances, the family Staphylinidae (Coleoptera) and the family Formicidae (Hymenoptera) were considered separately. Considering the limited number of individuals and the single-season sampling, we simply describe the seasonal diet of this population, comparing this information with those available for the population from the artificial spring [26], avoiding performing weak statistical analyses. The full data related to salamanders’ stomach contents is provided as Appendix A.

## 3. Results

During the first long-term monitoring, we performed a total of 929 observations of *Speleomantes sarrabusensis* within the two artificial springs (737 in the first and 102 in the second). A detailed summary of the observed individuals is shown in Table 1.

**Table 1 animals-13-00391-t001:** Monthly data from the long-term monitoring of *Speleomantes sarrabusensis*. In the upper part of the table, we show the monthly count performed within the two artificial springs coded following Lunghi, et al. [27]. The symbol (-) indicates that the survey has not been performed. In the lower part, we show the average monthly microclimatic conditions recorded inside the two artificial springs during the monitoring period. For each month, we show the average (±SD) temperature (°C) and relative humidity (%) recorded during the period 2015–2018. When the standard deviation is not shown, it means that the relative microclimatic variable has been recorded only once.

		Number of Observed Individuals											
Site	Year	January	February	March	April	May	June	July	August	September	October	November	December
Spring 1	2015	-	-	4	40	72	79	66	50	83	16	1	-
	2016	3	-	10	70	84	90	36	39	42	22	0	1
	2017	1	1	5	29	18	23	36	33	20	35	2	1
	2018	2	6	9	21	39	59	-	-	-	-	-	-
Spring 2	2015	-	-	0	6	10	2	5	5	7	18	3	-
	2016	3	-	8	8	5	3	3	1	1	4	5	3
	2017	2	2	8	12	6	5	6	5	3	7	5	3
	2018	3	5	6	0	8	6	-	-	-	-	-	-
		Data on the inner microclimate											
Site		January	February	March	April	May	June	July	August	September	October	November	December
Spring 1	Temperature	10.2 (±0.8)	10.3 (±0.1)	10.3 (±0.3)	12.2 (±1.3)	13.2 (±1)	14.2	16.4 (±0.8)	17.3 (±0.8)	16.5	15.2 (±0.3)	13.0 (±0.7)	9.8
	Humidity	81.2 (±2.2)	81.8 (±0.3)	82.7 (±0.9)	83.4 (±2.1)	84.1 (±3)	81.0	78.8 (±1.4)	79.7 (±2.6)	79.5	82.2 (±1)	82.4 (±1.5)	81.6
Spring 2	Temperature	10.0 (±1.3)	9.9 (±0.3)	10.9 (±0.1)	11.9 (±0.8)	12.6 (±0.7)	13.9	15.4 (±0.6)	16.0 (±0.9)	15.7	14.2 (±0.1)	12.8 (±1.3)	9.3
	Humidity	82.7 (±1)	82.6 (±1.1)	82.6 (±0.9)	84.1 (±2.8)	84.5 (±3.4)	82.6	82.2 (±0.4)	81.8 (±1)	81.6	82.9 (±1.2)	80.0 (±2.8)	82.4

The number of observed individuals was significantly affected by both the month (*F_11,61_* = 3.17, *p* = 0.002) and the year of the survey (*F_1,61_* = 4.72, *p* = 0.034). Overall, in June, the number of observed individuals was the highest, while in November, it was the lowest. The number of individuals significantly declined over the monitoring period. In Table 1 are also shown data on the monthly average (±SD) of the air temperature and humidity recorded within the two springs.

With the second part of our study, we were able to provide the first estimation of the abundance for four different populations of *S. sarrabusensis* (Table 2).

The species detection probability was relatively low (0.328 ± 0.081 SE). For the population inhabiting the two artificial springs on Sette Fratelli mountain, we estimated 34 individuals, while the estimated abundance for the other three epigean populations was 41, 9, and 9, respectively (Table 2).

In site forest #1 we captured a total of 37 individuals (24 + 15) of which 11 were males, seven females, and 21 juveniles (Appendix A). Six juveniles were too small and we did not perform stomach flushing; among the other individuals, none had an empty stomach. We recognised a total of 356 prey items belonging to 31 different prey categories; three categories accounted for more than 43% of the consumed prey (Entomobryomorpha, 25%; Araneae and Diptera both >9%) (Appendix A).

## 4. Discussion

With this study we provided the results of the first monitoring performed on *Speleomantes sarrabusensis*. Our results provided quantitative information on the monitored populations, data that represents the starting point for future conservation assessments. During the systematic monitoring performed in this study (2015–2018), we noticed a significant decline in the number of active individuals inside the springs. This can be considered just a red flag that should stimulate further monitoring because we have no additional information that helps us in understanding the potential causes. According to the data on the microclimate recorded inside the springs throughout the monitoring period (Table 1), we can exclude a drastic change in the inner microclimatic conditions as a potential cause for the reduction of the number of active individuals [8]. Indeed, from our data we can notice a yearly natural fluctuation of the inner microclimate correlated to the external climatic conditions [9]. Furthermore, when averaging the monthly data of the inner microclimate recorded throughout the survey period, we obtain a standard deviation whose maximum reaches 1.3 °C and 3.4% for temperature and humidity, respectively, a condition that maintains the inner microclimate within the preferred range for the species [4]. It is possible that some human-induced factors negatively affected the populations causing a sensible decline, but we also cannot exclude the possibility that we simply recorded a natural contraction of the population abundance; further monitoring activities are therefore necessary to better comprehend the causes of such decline.

In the second part of this study we provided the first quantitative information on the abundance of different populations of *S. sarrabusensis*. However, we must say that the observed individuals and the related estimation of abundance may be lower than expected. According to the studies performed on different subterranean populations of *Speleomantes*, during the period April–May, the number of active individuals was the highest [8,9,19], a condition that allows to produce estimations that are the closest to the real population size [22]. Considering the fully epigean habits of *S. sarrabusensis*, we chose to perform our surveys within the forested area in March, a month in which we expected suitable environmental conditions [8]. Unfortunately, the microclimatic conditions registered at the sites during our surveys were sub-optimal for the species [4] (Table 2), a circumstance that strongly affects the activity of individuals [8,9]. Therefore, our data should be interpreted as a snapshot taken in a sub-optimal period when the number of active individuals was very low, producing a highly underestimated estimation. Similarly, the surveys within artificial springs were performed in June, a period in which the subterranean activity of *Speleomantes* is the highest [4]. However, the extremely harsh climatic conditions occurred in that period likely induced an earlier aestivation of the population [1], allowing us to observe only a small number of active individuals, and therefore produced unexpected underestimated abundances. 

The trophic spectrum of this population was larger than that of the population from artificial springs (31 vs. 19 recognized prey categories) [26], suggesting that resource availability may play an important role in defining the diet and the foraging behaviour of these generalist salamanders [13,28,29]. According to the optimal forage theory, individuals should adopt a foraging strategy that maximises their food intake with the least effort [30]. Aggregative behaviour is often adopted by prey to dilute predation risk but, at the same time, it may allow predators to easily locate a very fruitful target. Considering the > 3000 prey items recognised from the *S. sarrabusensis* population occurring in the artificial spring [26], it can be noted that about 94% of the consumed prey belongs to just two categories: Diptera (76,45%) and Coleoptera Staphylinidae (17.42%). These relatively small prey are often observed in very dense clumps inside those artificial springs, giving insights on the reasons promoting the poorly diverse diet of salamanders occurring therein [13]. Indeed, if we only consider individuals that consumed these two types of prey, we observe that the average number (±SD) of consumed prey is 20.36 ± 25.29 for Diptera and 10.62 ± 14.3 for Coleoptera Staphylinidae. On the other hand, the individuals from the epigean population studied here showed more variability in their diet (Appendix A), and the most consumed prey was Entomobryomorpha, which represents the most diverse group of Collembola [31]. Although these Collembola can reach high densities [up to 1800 individuals per dm^3^; 31], in our case, each individual did not consume a very high number of Entomobryomorpha (3.42 ± 2.96), meaning that these prey may be locally common but do not show particularly gregarious behaviour.

## 5. Conclusions

Our study provided new information on the poorly known *Speleomantes sarrabusensis* through standardized methodologies that should be systematically adopted in the future to monitor the conservation status of this endangered amphibian species. Monitoring of epigean populations of *Speleomantes* is more challenging compared to subterranean ones [9,11,19], but they also require more attention as they have fewer opportunities to oppose human-induced alterations of the environment [32,33]. Furthermore, monitoring both epigean and hypogean populations may allow us to discover potential divergences in life traits between conspecific populations [34]. In conclusion, *S. sarrabusensis* needs particular attention and we are willing to continue with our data collection to provide important information useful for the conservation of this highly endangered species.

## Figures and Tables

**Figure 1 animals-13-00391-f001:**
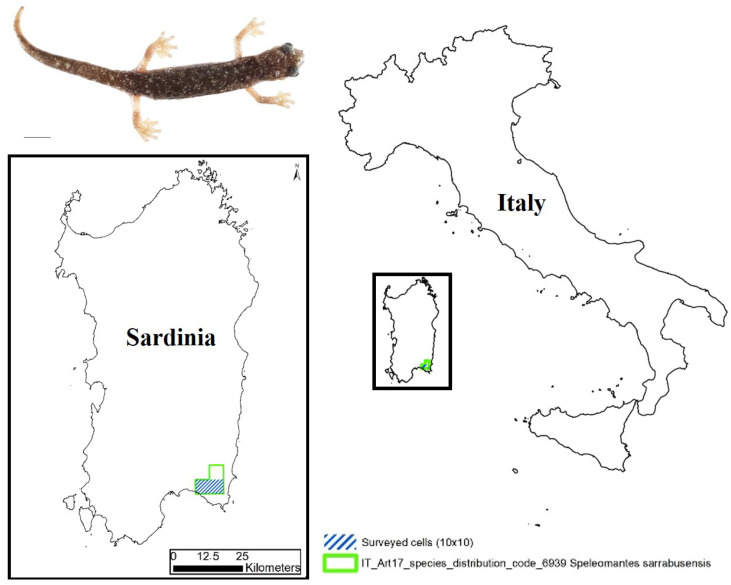
Map indicating the area of interest in our monitoring activity. In the upper-left corner, an adult of *Speleomantes sarrabusensis* (scale bar = 10 mm).

**Figure 2 animals-13-00391-f002:**
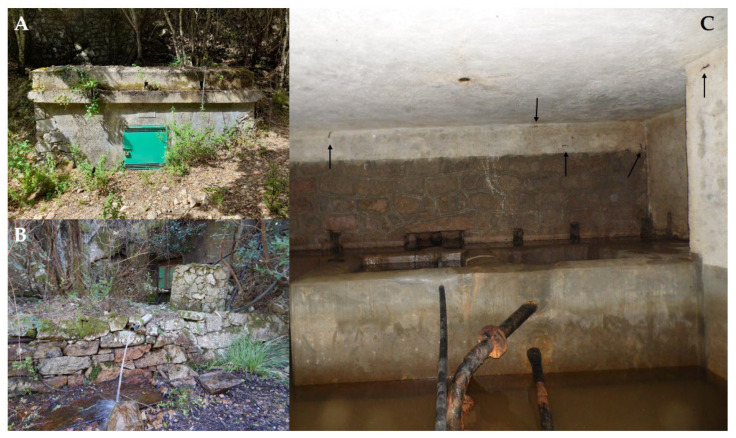
(**A**,**B**) show the two artificial springs in which *Speleomantes sarrabusensis* occurs. In (**C**), it is possible to see part of the inner environment of one spring; arrows are pointing to a few individuals that are clinging on the concrete walls.

**Table 2 animals-13-00391-t002:** Data related to the monitored populations of *Speleomantes sarrabusensis*. Along with general information on the location of the site, the surveyed areas, and the local microclimate, for each population we provide the number of observed individuals during each survey and the estimation of the population abundance. The elevation represents the average elevation of the monitored area. The abundance (mode) is accompanied by the confidence interval (2.5–97.5%).

Population	Latitude	Longitude	Elevation (m a.s.l.)	Surveyed Area (m^2^)	Microclimate	Survey #1	Survey #2	Survey #3	Estimated Abundance	2.5%	97.5%
Springs	39°29′	9°44′	783	70	15.4 °C—79.2%	21	14	3	34	28	41
Forest #1	39°12′	9°28′	444	2454	10.3 °C—80%	10	12	28	41	35	48
Forest #2	39°14′	9°28′	569	2628	10.3 °C—80.4%	2	0	1	9	5	16
Scrub #1	39°15′	9°23′	620	320	8.7 °C—79.9%	0	2	0	9	4	15

## Data Availability

Data for *Speleomantes sarrabusensis* stomach contents are provided as Appendix A.

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
