# Peer review of "Monitoring of the Endangered Cave Salamander Speleomantes sarrabusensis"

_animals, 2023, doi:10.3390/ani13030391_

Round 1

Reviewer 1 Report

In this manuscript authors present results of monitoring populations in artificial springs and those in forested area of European plethodonid salamanders Speleomantes sarrabusensis, by recording the number of individuals and temperature, and humidity.

The first effort to monitor the endangered species of European plethodonid salamanders is valuable and appreciated.

Author Response

Thanks for positive comments

Reviewer 2 Report

The paper is well written and clearly presented. I have mainly highlighted some minor points of grammar.

line 17: schemes, no scheme

25: better to phrase as 'surveyed for populations at three time points'

36: Sardegna in text, Sardinia in Fig 1. I'm sure either will be fine but better to use the same language for both

Fig 1: can the authors add a scale bar to the image of the salamander?

50: poorest, not poorer

56: space in which to hide

58: can the authors briefly describe the 'artificial springs', to give the reader a better idea of this habitat?

61: knowledge of

74: need for, not need of

84: citation for the linear model used could be added here

211: the 19 categories of prey in the artificial spring population are not referred to in the results section - please cite the reference here if this is from previously published work

216: >3000 prey - do you mean >3000 prey items?

225: table S1 is not included in the review MS

Reviewer 3 Report

This is an important contribution to our understanding of this unusual and imperiled salamander.  The authors do an excellent job of citing relevant European studies.  There is a decent body of similar studies on North American plethodontids. See Taylor, et al., 2015 for a study on spring (surface) and cave populations of Plethodon on Fort Hood, Texas.  

Maps (plan and profile) and photos of the springs and surface sites would be helpful. 

Is the footprint of the springs 70 m2 or is that the total surface area (including areas of walls)?   A heatmap or other showing the location of salamanders would be helpful in deciphering observed patterns.  Cave surveys are notoriously difficult because the amount of humanly-inaccessible habitat is often extremely small relative to the total amount of habitat available.  Of course, this is also modulated by climate and seasonal variation.  

More detail regarding surface surveys would be helpful.  What is the difference in search hours/m2 among the sites?  Was  GPS tracking used to ensure that the 320 m2 plot was adequately surveyed?  Are areas of dense shrubs simply difficult to access for researchers or are they non-habitat?

Is it possible to access bioclimatic data for the area to assess long term and episodic trends that may influence apparent salamander abundance?
